

# Automatic detection of the parasite *Trypanosoma cruzi* in blood smears using a machine learning approach applied to mobile phone images

Mauro César Cafundó Morais[1,2,3,*], Diogo Silva[3,*], Matheus Marques Milagre[4], Maykon Tavares de Oliveira[6], Thaís Pereira[5], João Santana Silva[6], Luciano da F. Costa[7], Paola Minoprio[2], Roberto Marcondes Cesar Junior[8], Ricardo Gazzinelli[5], Marta de Lana[4,9] and Helder I. Nakaya[1,2,3,10]

[1] Hospital Israelita Albert Einstein, São Paulo, Brazil
[2] Scientific Platform Pasteur-University of São Paulo (SPPU), Universidade de São Paulo, Sao Paulo, SP, Brazil
[3] Department of Clinical and Toxicological Analysis, School of Pharmaceutical Sciences, Universidade de São Paulo, Sao Paulo, SP, Brazil
[4] Departamento de Análises Clínicas (DEACL), Programa de Pós-graduação em Ciências Farmacêuticas (CiPHARMA), Universidade Federal de Ouro Preto, Ouro Preto, MG, Brazil
[5] Laboratório de Imunopatologia, Instituto René Rachou, Fundação Oswaldo Cruz, Universidade Federal de Minas Gerais, Belo Horizonte, MG, Brazil
[6] Fiocruz- Bi-Institutional Translational Medicine Project, FIOCRUZ/SP, Ribeirão Preto, SP, Brazil
[7] São Carlos Institute of Physics (DFCM- IFSC), Universidade de São Paulo, São Carlos, SP, Brazil
[8] Instituto de Matemática e Estatística (IME), Universidade de São Paulo, São Paulo, SP, Brazil
[9] Núcleo de Pesquisas em Ciências Biológicas (NUPEB), Universidade Federal de Ouro Preto, Ouro Preto, MG, Brazil
[10] Center of Research in Inflammatory Diseases (CRID), Universidade de São Paulo, Ribeirão Preto, SP, Brazil
[*] These authors contributed equally to this work.

Corresponding authors
Mauro César Cafundó Morais,
mauro_morais@usp.br,
mauroccm@gmail.com
Helder I. Nakaya, hnakaya@usp.br

## ABSTRACT

Chagas disease is a life-threatening illness caused by the parasite *Trypanosoma cruzi*. The diagnosis of the acute form of the disease is performed by trained microscopists who detect parasites in blood smear samples. Since this method requires a dedicated high-resolution camera system attached to the microscope, the diagnostic method is more expensive and often prohibitive for low-income settings. Here, we present a machine learning approach based on a random forest (RF) algorithm for the detection and counting of *T. cruzi* trypomastigotes in mobile phone images. We analyzed micrographs of blood smear samples that were acquired using a mobile device camera capable of capturing images in a resolution of 12 megapixels. We extracted a set of features that describe morphometric parameters (geometry and curvature), as well as color, and texture measurements of 1,314 parasites. The features were divided into train and test sets (4:1) and classified using the RF algorithm. The values of precision, sensitivity, and area under the receiver operating characteristic (ROC) curve of the proposed method were 87.6%, 90.5%, and 0.942, respectively. Automating image analysis acquired with a mobile device is a viable alternative for reducing costs and gaining efficiency in the use of the optical microscope.

## INTRODUCTION

Chagas disease is a life-threatening illness caused by infection with the protozoan *Trypanosoma cruzi* (*Chagas, 1909*). Most of the cases occur when metacyclic trypomastigotes eliminated in the feces or urine of the vector enter the human host. Infection may also develop through oral ingestion of contaminated food, congenital transmission, blood transfusion, transplants of organs, and laboratory accidents (*Cancino-Faure et al., 2015*; *Filigheddu, Górgolas & Ramos, 2017*; *Luquetti et al., 2015*). After penetrating the host cells, the metacyclic trypomastigotes differentiate into amastigote forms in the cytoplasm. Subsequently, the amastigotes multiply themselves by binary fission and transform into trypomastigotes that disrupt the host cell, releasing into the bloodstream. These circulating trypomastigotes may invade other host cells or be ingested by vectors (*Lana & Machado, 2017*).

The acute phase of Chagas disease is characterized by high parasitemia in the blood (*Dias et al., 2016*; *Luquetti & Schmuñis, 2017*). This allows the visualization of bloodstream forms in the blood of infected individuals using a parasitological fresh-blood test, as well as smear and thick drop blood tests (*Gomes, Lorena & Luquetti, 2009*). Laboratory diagnosis, however, has a few key limitations. First, it should be performed by trained microscopists who observe the parasites. Reliance on professionals with various skills makes the diagnosis prone to errors and heterogeneous. Second, manual search and detection of parasites is a laborious task. This often delays the laboratory result, which further delays the initiation of treatment. Third, methods that improve the search for the parasite in microscopic images, such as attaching a dedicated high-resolution camera system to the microscope, are usually expensive and often prohibitive for low-income settings.

Machine learning (ML) algorithms can assist in the laboratory diagnosis of acute Chagas disease. They can automate the search and detection of the parasites, improving the reproducibility of image analysis. These algorithms have been applied to image detection of *T. cruzi* as well as other parasites that circulate in the blood (*Rajaraman et al., 2018*; *Uc-Cetina, Brito-Loeza & Ruiz-Piña, 2015*; *Górriz et al., 2018*). Previous work relied on an image acquisition system with a dedicated camera attached to the microscope for the detection of *T. cruzi*. This system is important for proper ML training since, it produces homogeneous images regarding the lightness, color, acquisition time, aperture, and resolution. However, these dedicated cameras are often expensive, which makes such system prohibitive to low-income settings. The use of mobile device cameras increases image diversity, which can make ML models even more robust. Models trained in mobile device imaging were developed for the detection of *Plasmodium ssp.* which causes malaria (*Rosado et al., 2016*; *Oliveira et al., 2017*; *Yang et al., 2020*). To date, models for the detection of *T. cruzi* in images acquired with mobile devices have not been developed.

Here, we developed and evaluated a ML algorithm that detects the bloodstream forms of *T. cruzi* based on features of segmented body from the parasites. The extracted features consisted in descriptors of morphology (geometric), color and texture, as well as statistical descriptors (Hu's invariant moments). We tested our method using images acquired with a mobile phone from the acute phase of infection in a murine model. Our model reached 89.5% accuracy in a set of images that were not previously presented in the training process. The proposed method presented an acceptable performance to detect trypomastigotes using mobile phone images of blood smear.

## MATERIALS & METHODS

### Samples analyzed

A total of 33 slides with thin blood smears of Swiss mice experimentally infected with *T. cruzi* Y strain at acute phase of infection were prepared for image annotation and analysis. Sample preparations were obtained from animals of the Laboratory of Chagas disease, Federal University of Ouro Preto where the *T. cruzi* strain was maintained through successive blood passages in mice. The Ethics Committee for the Use of Animals (CEUA) at the Federal University of Ouro Preto, Laboratory of Chagas Disease, Minas Gerais, Brazil provided approval for this research (CEUA No. 2015/50).

### Object segmentation

We standardized the resolution of the images to $768 \times 1,024$ pixels$^2$ before segmenting the parasites. We applied a graph-based segmentation method (*Felzenszwalb & Huttenlocher, 2004*). In this process, a graph-based representation of the image is defined were pixels corresponds to vertices and neighboring pixels are connected edges. The contours of the regions of interest with the parasites are obtained by selecting the edges between the different regions based on the differences in intensity between the regions, and the difference in intensity between the pixels within each region. As a result, we observed the whole parasite cell body segmented within each region.

Next, we cropped a $100 \times 100$ pixels$^2$ region around each parasite based on manual position annotation. Only the regions of interest with the parasite were selected for processing and feature extraction. This procedure resulted in 1,314 parasites. We selected other segments from the images with features very similar to the parasite using nearest neighbors technique. In other words, we selected objects that do not belong to parasites and clustered under the label "Unknown". In this way, we were able to obtain the same number of objects (*T. cruzi* or unknown) for the object classification task.

### Feature extraction

After the segmentation of the parasites, we performed the conversion from the RGB color space to CIEL*a*b* color space for object's feature extraction (see the Supplemental Information file for details). These features are object descriptors and are classified as geometric (perimeter, area, circularity, thickness ratio, centroid to contour distances, major and minor axis, aspect ratio, etc.); curvature (entropy, bending energy, standard deviation, and variance) (*Costa & Cesar Jr, 2009*); texture (color co-occurrence matrices,

entropy, inverse difference moment, angular second moment, contrast, correlation) color (mean, median, mode, amplitude, and variance) (*Gui et al., 2013*; *Palm, 2004*) descriptors and Hu's invariant moments (*Huang & Leng, 2010*).

## Feature selection

To increase performance, reduce the noise, and avoid overfitting, we applied either Principal Component Analysis (PCA). In this manner, we reduced the number of features used by the algorithm during the training. The dimension reduction with PCA was performed by keeping the 16 highest eigenvalues of the covariance matrix derived from the feature space data. The proportion of variance of the 16 principal components correspond to 95% of the original variance. Therefore, we obtained a new feature space data that consisted of eigenvectors of the principal components. We obtained the final dataset by multiplying the transposed matrix of eigenvectors by the input data matrix centered by the mean.

## Object classification

We applied supervised learning classification with support vector machines (SVM), k-Nearest Neighbours (KNN) and Random Forest (RF) as they presents good generalization with a small dataset (*Chen et al., 2020*; *Hsu, Chang & Lin, 2016*; *Ben-Hur & Weston, 2010*; *Cunningham & Delany, 2007*; *Breiman, 2001*).

In the SVM method, each sample is represented by a point in an $n$-dimensional space, where $n$ is the number of object features and its values are the coordinates of the point, including its class. The classifier searches for the optimal hyperplane that will be used to split the points belonging to distinct classes. In the KNN method, a sample is classified to the data label which has the most representatives within the $k$ nearest neighbors features of the sample. In the RF method, the classification of each object happens through a combination of decision trees. The classification performance of each tree in the ensemble is improved by bootstrapping sampling and aggregation from the training set. The final classification is made by averaging each decision tree probabilistic prediction. In addition, we also performed classification task using an ensemble of these methods based on "soft" voting classifier. The voting classifier combines the results of the different predictors and use the average of predicted probabilities to assign the class labels to each sample.

We used Python's scikit-learn library to train and validate the models (*Pedregosa et al., 2011*). We performed object feature data standardization before classification since data presented different orders of magnitude. We assessed the classification performance of the methods by comparing the samples in the feature space for a given class with all other samples. After finding the highest accuracy value for each model, we assessed the models in the validation set.

## Statistical analysis

The performance of the constructed models was assessed by Receiver Operating Characteristic (ROC) curve analysis, where the average of the area under the curve (AUC) was calculated for the quantification in both training set and validation sets. We also evaluated the model performance based on sensitivity, specificity, precision, and F1-score.

A. Sample preparation

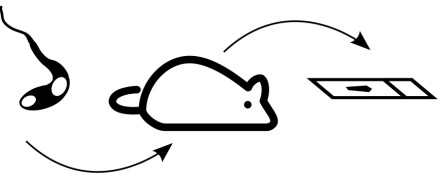

B. Image acquisition

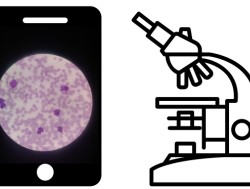

C. Parasite labeling and Image processing

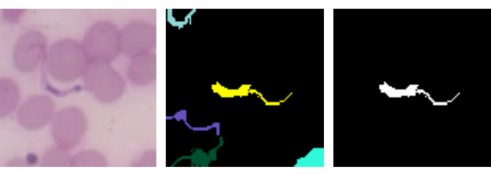

D. Feature extraction and selection

Geometric
Color
Texture
Curvature
Invariant moments

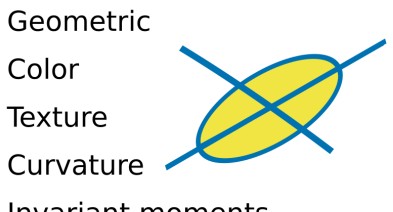

E. Machine learning model training and validation

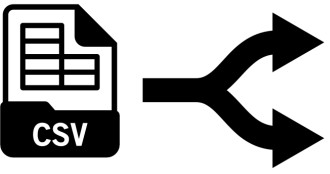

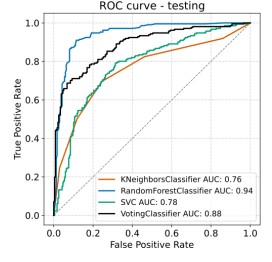

F. Parasite detection

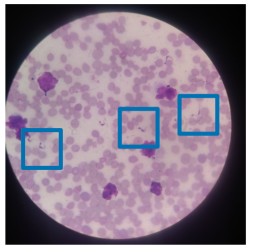

**Figure 1** ***T. cruzi* detection image analysis pipeline.** (A) Blood smear samples were prepared from mice experimentally infected with *T. cruzi* parasites at the acute infection stage. (B) Images of thin blood smear slides were acquired with a mobile phone camera attached to a microscope ocular lens. (C) Parasite (trypomatigote forms of *T. cruzi*) was segmented by a graph-based algorithm. (D) Images were converted to CIEL\*a\*b\* color space and parasite features were extracted and selected (PCA). (E) Objects feature data were split into training and test sets. Four machine learning models were trained and assessed. (F) Parasites were detected in mobile phone camera images.

# RESULTS

## Image analysis pipeline for detecting *T. cruzi*

We developed and tested an image analysis pipeline which was based on a ML model to detect *T. cruzi* in the blood during acute infection (Fig. 1). Initially, the images were prepared from blood samples collected from female Swiss mice acutely infected with the Y strain of *Trypanosoma cruzi*. The collected blood sample smears were stained by the Giemsa method (*Vallada, 1999*). This technique allows the observation of parasites with oil immersion objectives. Regions of nucleic acid-rich in adenine-thymine bond tend to get darker. On the other hand, regions rich in cytosine-guanine bonds are less prone to embody the Giemsa stain and tend to present clearer stains.

A total of 33 slides with thin blood smears from different animals were stained with Giemsa. About 20 fields of view that correspond to each of the analyzed images were captured from each slide. We manually acquired 674 images from the slides under 100x objective (CFI E Plan Achromat 100x Oil, 1.25 NA/0.23 W.D.) in an optical microscope (Nikon Eclipse E200) with a cell phone camera (Morola Moto G4) attached to the eyepiece (CFI E 10x, F.N. 20 mm) (Fig. 2). The camera was configured with the macro focus and other configurations were set to automatic for acquisition. With these settings, the images were acquired with a resolution of 3,456 × 4,608 pixels$^2$, a field of view with a diameter

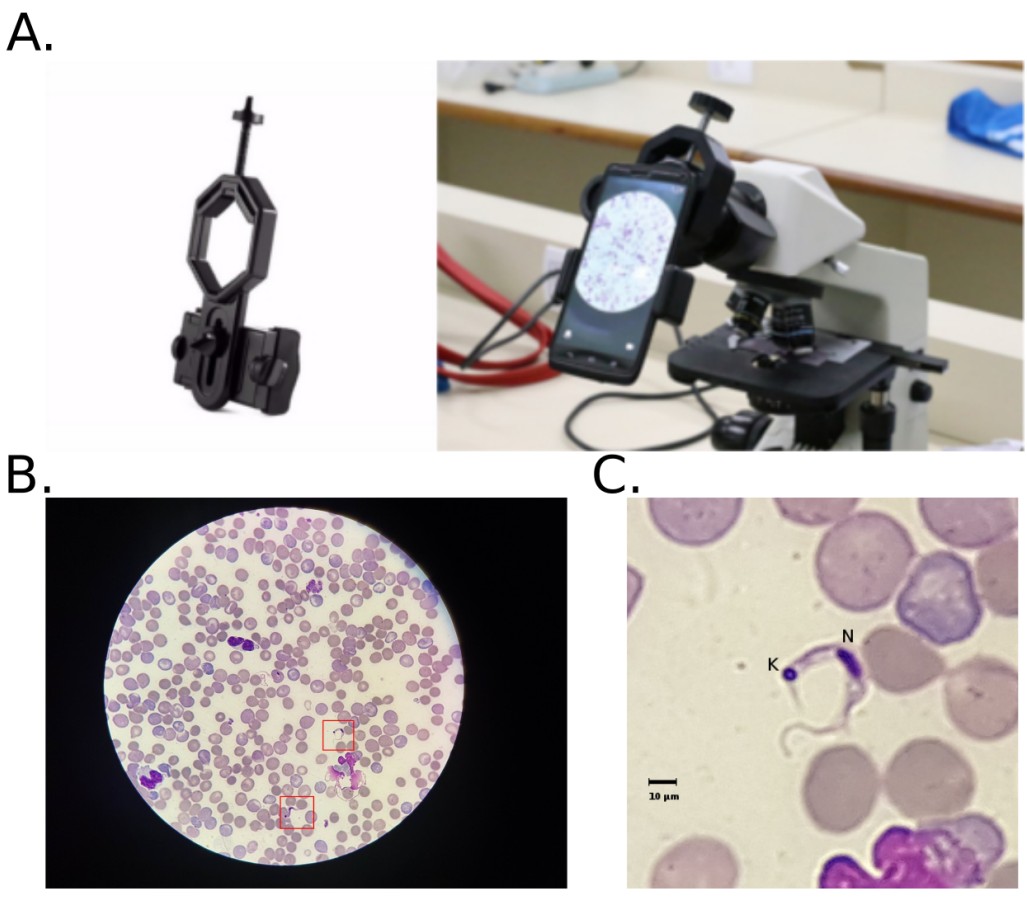

**Figure 2 Mobile phone attached to optical microscope objective lens and image acquisitio.** (A) The mobile phone was attached to the microscope ocular lens (eyepiece) with a plastic support device (left). The camera was configured with the macro focus for acquisition. Other configurations were set to automatic. (B) Image of field-of-view from blood smear slide of mice infected with *Trypanosoma cruzi*. The red squares indicate regions with the presence of a parasite. (C) Crop of a field of view with the *T. cruzi* parasite at the center. K, kinetoplast; N, nucleus; Scale bar, 10 μm.

of 0.2 mm resulted in a pixel lateral size of approximately 0.06 μm. A few images were acquired at $2,448 \times 3,264$ pixels$^2$ due to change in camera configuration. This was useful for classifier test at different input image resolutions. Images were stored in JPEG format, at 100% quality, and file names standardized according to unique identifiers.

Parasites were microscopically identified by two specialist researchers in *T. cruzi*. A total of 1,314 parasites were observed. We then marked the position of each nucleus or cell body of the *T. cruzi* found in the images. The position of the objects of interest (parasite) in the image was obtained by a point-and-click event using an "in-house software". The image identifier, the pointed object, and the coordinates in *X* and *Y* axis information were extracted and stored in a database (Data S1).

To detect the objects of interest, we applied a graph-based approach (*Felzenszwalb & Huttenlocher, 2004*). Segmented regions with more than 3,000 pixels were considered not to contain the objects of interest. After image segmentation, objects were cropped in a 100 ×

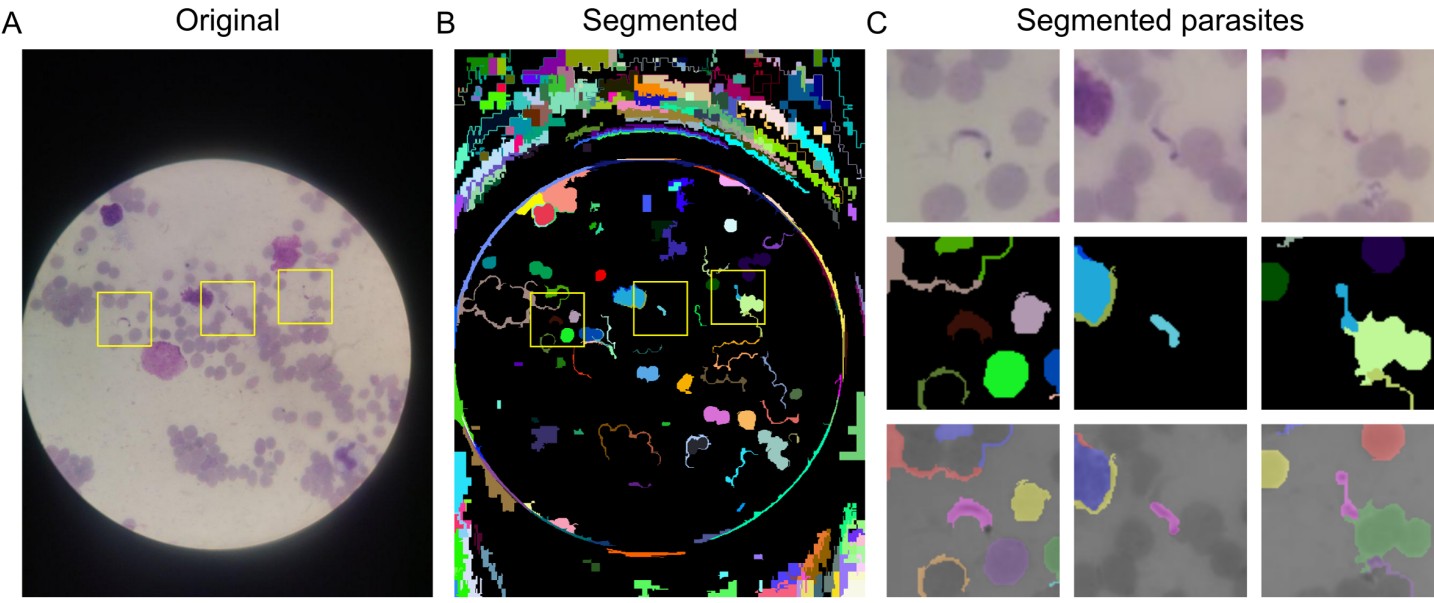

**Figure 3** **Object segmentation.** (A) Original image acquired with a mobile phone attached to the microscope. (B) Segmented image with regions highlighted with different colors. Yellow squares indicate the location of the parasite. (C) Segmented parasites in a $100 \times 100$ pixels$^2$. Top row: *T. cruzi* trypomastigotes from orignal image. Middle row: segmented regions with parasites. Bottom row: segmented parasites within the segmented region of interest highlighted. Only the regions segmented with the parasites were selected for feature extraction.

100 pixel$^2$ sub-region around its $X$ and $Y$ coordinates and labeled into two classes: parasite and unknown (Fig. 3). The segmented regions containing the the *T. cruzi* trypomastigote form were labeled as "parasite" (Fig. 4A). The segmented regions that do not contain a parasite or that are over-segmented were labeled as "unknown" (Fig. 4B).

We selected a set of regions labeled as "unknown" to train and validate the classifier method. The regions were selected based on the features values closest to the regions labeled as "parasite" using the nearest neighbors method. In this way we achieved a total of 1,314 segments marked as parasite and the same number of segments marked as unknown (Table 1).

To identify the parasite in the image sub-region, we first extracted features from these regions. These features represent a description of the object's morphology (geometry and curvature), as well as color and texture. We also calculated Hu's invariant moments to capture information regarding shape and intensity regardless of the object's position and size (Table 2). In total, we extracted 49 features from the segmented objects of each class (Data S2).

We then split the region's feature data into two sets based on the number of acquired images. This resulted in a proportion of about 80% of regions for the training set ($n = 2181$) and 20% for test ($n = 447$). We also applied principal component analysis (PCA) to reduce the number of features and test whether it may improve the classification performance. The proportion of variance of the 16 principal components corresponds to 95% of the original variance of the data (Fig. S1). Therefore, we used 16 features of the transformed values matrix to train the model.

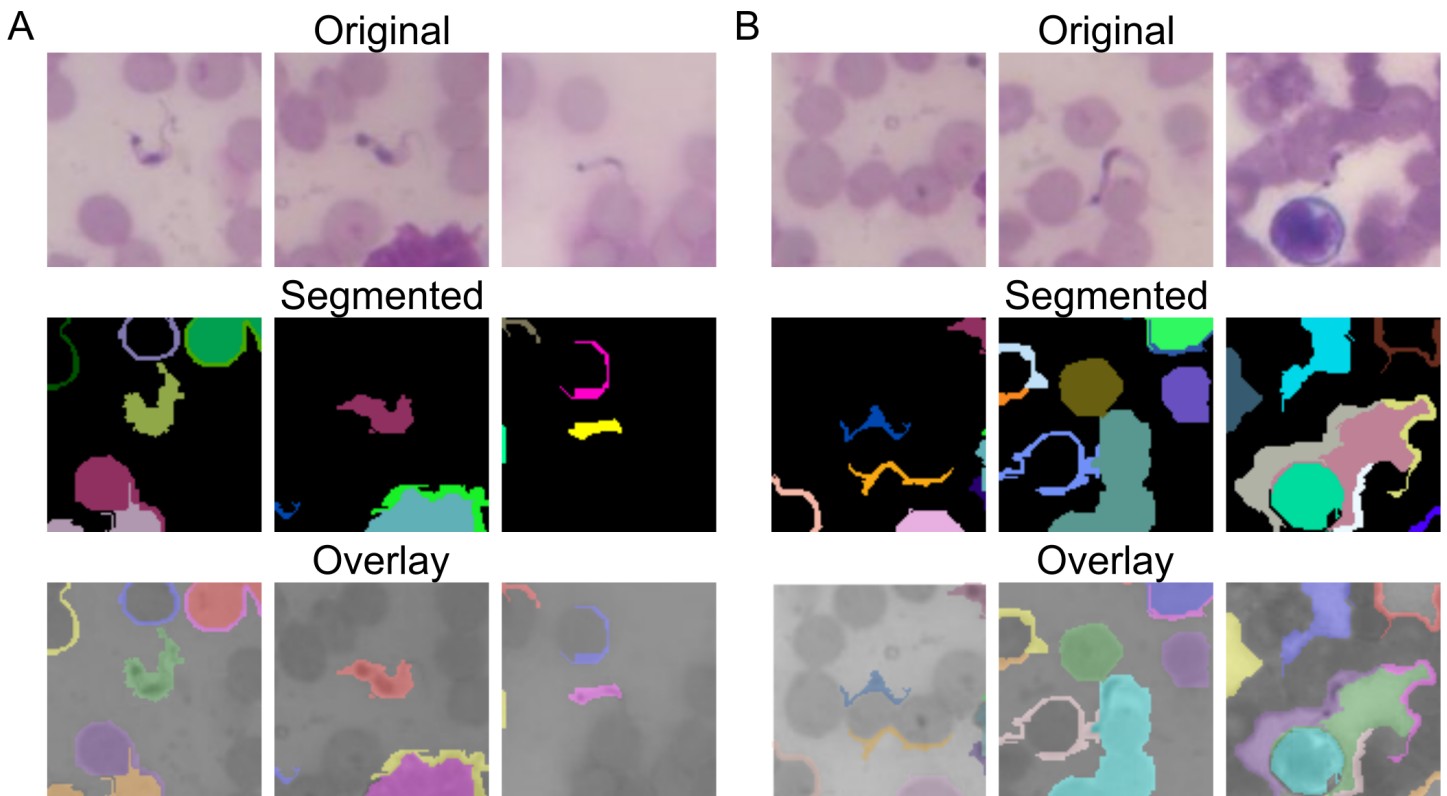

**Figure 4** **Parasite's segmentation and region labeling.** (A) Example of segmented regions that contain the parasite. The segmented regions containing the *T. cruzi* trypomastigote form were labeled as "parasite". (B) The segmented regions that do not contain a parasite or that are over-segmented were labeled as "unknown".

**Table 1** **The number of objects by classes used in the training and test sets.**

| Class | Training | Test |
|---|---|---|
| Parasites | 1103 | 211 |
| Unknown | 1078 | 236 |
| Total | 2181 | 447 |

## Object classification task presented acceptable performance

We developed a classifier algorithm based on the object features. We observed better performance in the classification task without the feature selection. The models trained on features data selected by PCA were not able to generalize well on the test set, indicating overfitting (Table S1). Without the feature selection, the random forest classifier model presented acceptable performance with an accuracy of 99.7% and area under the ROC curve of of 1.0 in the training set, all the while presenting accuracy of 89.5% and AUC of 0.942 in the test set (Table 3). The voting classifier presented accuracy of 93.3% and AUC of 0.978 in the training set, and accuracy of 79.4% and AUC of 0.884 in the test set (Fig. 5).

The voting classifier confusion matrix presented sensitivity and specificity values of 76.8% and 81.8%, respectively (Table 4). The lower performance presented by the

Morais et al. (2022), *PeerJ*, DOI 10.7717/peerj.13470

Peerj

**Table 2  Object feature metrics.**

| Feature | Description | References |
|---|---|---|
| **Geometric** | | *Costa & Cesar (2009)* |
| Perimeter (P) | Parametric representation of the contour and its points identified by the coordinates $x(t)$ and $y(t)$ | |
| Area (A) | Integral of the contour | |
| Area and perimeter ratio | $\frac{A}{P}$ | |
| Circularity | $4\pi \frac{A}{P^2}$ | |
| Thickness ratio | $\frac{P^2}{A}$ | |
| Centroid | Given the center of mass M of a contour of complex signal $u(n)$, the centroid coordinates $(z_1, z_2)$ were obtained by the average of all the points in $u(n)$. | |
| Centroid to contour maximum distance | The distance between the centroid and the furthest point on the contour. | |
| Centroid to contour minimal distance | The distance between the centroid and the nearest point on the contour. | |
| Centroid to contour average distance | The average of the distances between the centroid and all points in the contour. | |
| Major axis | Pair of more distant points belonging to the object. | |
| Minor axis | Pair of closest points belonging to the object. | |
| Aspect ratio | $\frac{Majoraxis}{Minoraxis}$ | |
| Perimeter and Major axis ratio | $\frac{P}{Majoraxis}$ | |
| Bilateral symmetry | Bilateral symmetry is given by the proportion of the number of pixels between the intersection of an object and it's reflecting shape with respect to the major axis, and the union between those two objects. | |
| **Hu's invariant moments**[a] | | *Hu (1962)*; *Huang & Leng (2010)* |
| $\phi_1$ | $\eta_{20} + \eta_{02}$ | |
| $\phi_2$ | $(\eta_{20} - \eta_{02})^2 + 4\eta_{11}^2$ | |
| $\phi_3$ | $(\eta|30 - 3\eta_{12})^2 + (3\eta|21 - \mu_{03})^2$ | |
| $\phi_4$ | $(\eta_{30} - \eta_{12})^2 + (\eta_{21} + \mu_{03})^2$ | |
| $\phi_5$ | $(\eta_{30} - 3\eta_{12})(\eta_{30} + \eta_{12})[(\eta_{30} + \eta_{12}) - 3(\eta_{21} + \eta_{03})^2] + (3\eta_{21} - \eta_{03})(\eta_{21} + \eta_{03})[3(\eta_{30} + \eta_{12})^2 - (\eta_{21} + \eta_{03})^2]$ | |
| $\phi_6$ | $(\eta_{20} - \eta_{02})[(\eta_{30} + \eta_{12})^2 - (\eta_{21} + \eta_{03})^2] + 4\eta_{11}(\eta_{30} + \eta_{12})(\eta_{21} - \eta_{03})$ | |
| $\phi_7$ | $(3\eta_{21} - \eta_{03})(\eta_{30} + \eta_{12})[(\eta_{30} + \eta_{12})^2 - 3(\eta_{21} - \eta_{03})^2] - (\eta_{30} - 3\eta_{12})(\eta_{21} + \eta_{03})[3(\eta_{30} + \eta_{12})^2 - (\eta_{21} + \eta_{03})^2]$ | |

Peer J

**Table 2** (*continued*)

| Feature | Description | References |
|---|---|---|
| **Color** | | *Burger & Burge (2016)* |
| Mean | $\frac{1}{MN}\sum_{i=1}^{M}\sum_{j=1}^{N}p_{ij}$ | |
| Median | $p_{(n+1)/2}$ | |
| Mode | Pixel value that occurs with greatest frequency | |
| Amplitude | max(p) - min(p) | |
| Variance | $\sum_{i=1}^{M}\sum_{j=1}^{N}p_{ij}\cdot(x_{ij}-\mu)^2$ | |
| **Curvature**[b] | | *Costa & Cesar (2009)* |
| Bending energy | $B=\frac{1}{P}\int k(t)^2 dt$ | |
| Variance | $Var(t)=\sum_{i=0}^{t}p_i\cdot(x_i-\mu)^2$ | |
| Entropy | $H(t)=-\sum_{i=0}^{t}p(k(t))\cdot\log(p(k(t)))$ | |
| **Color texture**[c] | | *Gui et al. (2013)* |
| Entropy (E) | $E=-\sum_{i=1}^{L}\sum_{j=1}^{L}p(i,j)\log(p(i,j))$ | |
| Angular second moment (ASM) | $ASM=\sum_{i=1}^{L}\sum_{j=1}^{L}(p(i,j))^2$ | |
| Contrast (CON) | $CON=\sum_{k=0}^{L-1}k^2(\sum_{|i-j|=k}p(i,j))$ | |
| Inverse difference moment (IDM) | $IDM=\frac{\sum_{i=1}^{L}\sum_{j=1}^{L}p(i,j)}{|1+i-j|}$ | |
| Correlation (COR) | $COR=\frac{\sum_{i=1}^{L}\sum_{j=1}^{L}(i,j)p(i,j)-\mu_x\mu_y}{\sigma^2}$ | |

**Notes.**

[a]Refer to *Huang & Leng (2010)* for $\mu$ and $\eta$ equations.

[b]The curvature k(t) of a parametric curve c(t) = (x(t), y(t)) was defined as $k(t)=\frac{x'(t)y''(t)-x''(t)y'(t)}{(x'(t)^2-y'(t)^2)^{3/2}}$, where $x'(t), y'(t)$ and $x''(t), y''(t)$ are the first and second derivative of the contour signal $x(t)$ and $y(t)$, respectively.

[c]Texture features were extracted based on the color co-occurrence matrix (CCM).

**Table 3  Prediction performance of models on the training and testing sets.**

| Set | Feature selection | Model | Metrics (%) | | | | | |
|---|---|---|---|---|---|---|---|---|
| | | | Sensitivity | Specificity | Precision | Accuracy | F$_1$-score | AUC |
| Train | None | SVM | 67.4 | 80.2 | 77.7 | 73.8 | 72.2 | 0.797 |
| | | KNN | 75.6 | 84.2 | 83.1 | 79.9 | 79.2 | 0.878 |
| | | RF | 99.8 | 99.5 | 99.5 | 99.7 | 99.7 | 1.0 |
| | | Ensemble | 88.6 | 92.0 | 91.9 | 90.3 | 90.2 | 0.978 |
| Test | None | SVM | 69.7 | 75.4 | 71.7 | 72.7 | 70.7 | 0.78 |
| | | KNN | 69.7 | 75.4 | 71.7 | 72.7 | 70.7 | 0.759 |
| | | RF | 90.5 | 88.6 | 87.6 | 89.5 | 89.0 | 0.942 |
| | | Ensemble | 76.8 | 81.8 | 79.0 | 79.4 | 77.9 | 0.884 |

**Notes.**

AUC, Area under the curve; SVM, Support vector machines; KNN, $k$-nearest neighbors; RF, random forest.

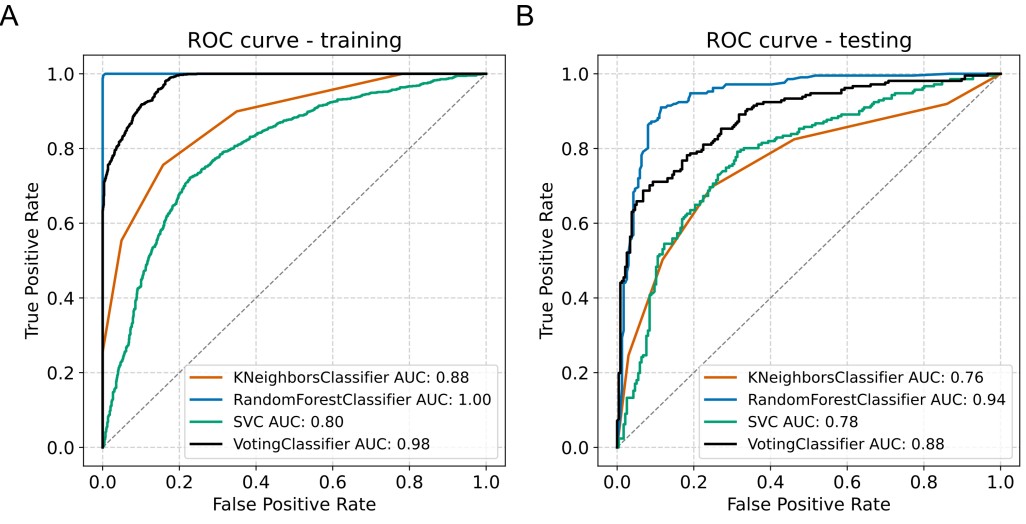

**Figure 5  Model classification performance.** (A) Receiver operating characteristic (ROC) curve in the training set. (B) ROC curve in the testing set. AUC, area under the curve.

voting classifier is because the ensemble's prediction is based on the average of the prediction probabilities of each classifier. Since the SVM and KNN classifiers presented lower performance individually (both presented sensitivity of 69.7% and specificity of 75.4%, Table 3) the prediction of the ensemble was lower. On the other hand, the RF classifier had a sensitivity of 90.5% and a specificity of 88.6% (Table 5).

The objects found in the images were then marked by the algorithm as parasite. We found two problems in the parasite recognition task. The first is related to the high rate of false positives (Fig. 6A). The regions of the images with leukocytes or high density of red cells showed over-stained areas. These areas were difficult to classify by the algorithm. The second problem was the false negative rate (Fig. 6B). Parasites in regions of the image that presented low contrast or low sharpness ("out of focus"), most commonly found at the

**Table 4** Confusion matrix of the voting classifier (ensemble's prediction) in the test set.

|  |  | True label | |
|  |  | Parasite | Unknown |
| --- | --- | --- | --- |
| Predicted label | Parasite | 162 | 43 |
|  | Unknown | 49 | 193 |

**Table 5** Confusion matrix of the Random Forest classification model in the test set.

|  |  | True label | |
|  |  | Parasite | Unknown |
| --- | --- | --- | --- |
| Predicted label | Parasite | 191 | 27 |
|  | Unknown | 20 | 207 |

edges of the field of view, were not recognized by the algorithm. This second problem is much more significant, since undiagnosed Chagas disease can put a person's life at risk.

## DISCUSSION AND CONCLUSIONS

In this work, we present an algorithm for automatic detection of the *T. cruzi* parasite in images acquired with a mobile phone device. Our approach involved image segmentation with a graph-based method, extraction of parasite features, selection of the most important features, and classification of these features with an Random Forest model for the detection of the parasite in the image.

The detection of *T. cruzi* in images was previously done using several classification models, such as gaussian discriminant, k-nearest neighbors, AdaBoost + SVM, and convolutional neural networks (CNN) (*Soberanis-Mukul et al., 2013*; *Uc-Cetina, Brito-Loeza & Ruiz-Piña, 2013*; *Uc-Cetina, Brito-Loeza & Ruiz-Piña, 2015*; *Pereira et al., 2020*). Despite these works reported a good performance (sensitivity and specificity >85%), all of them made use of images acquired with a dedicated camera system. Our method obtained a sensitivity of 90.5% and a specificity of 88.6% even though we used images with lower resolution (less than 1 megapixel). Machine learning approaches applied to images obtained from mobile device cameras showed similar performance (sensitivity of 80.5% and specificity of 93.8%) in detecting the malaria agent *Plasmodium* spp. (*Oliveira et al., 2017*; *Rosado et al., 2016*). Therefore, our method was the first to combine machine learning algorithms and low-resolution images to automatically detect *T. cruzi* parasite (Table 6).

We can enhance the classification task by testing other models. Currently, one of the techniques most used in pattern recognition are deep learning approaches (*Acevedo et al., 2019*; *Moen et al., 2019*; *Schmidhuber, 2015*). Deep learning approaches presented better performance in detecting *Plasmodium* ssp.(sensitivity of 94.5% and specificity of 96.9%) (*Rajaraman et al., 2018*). However, to build an effective model using these techniques to detect *T. cruzi*, huge data sets are required where the performance of the model increases in logarithmic proportion to the volume of images (*Sun et al., 2017*). Another challenge

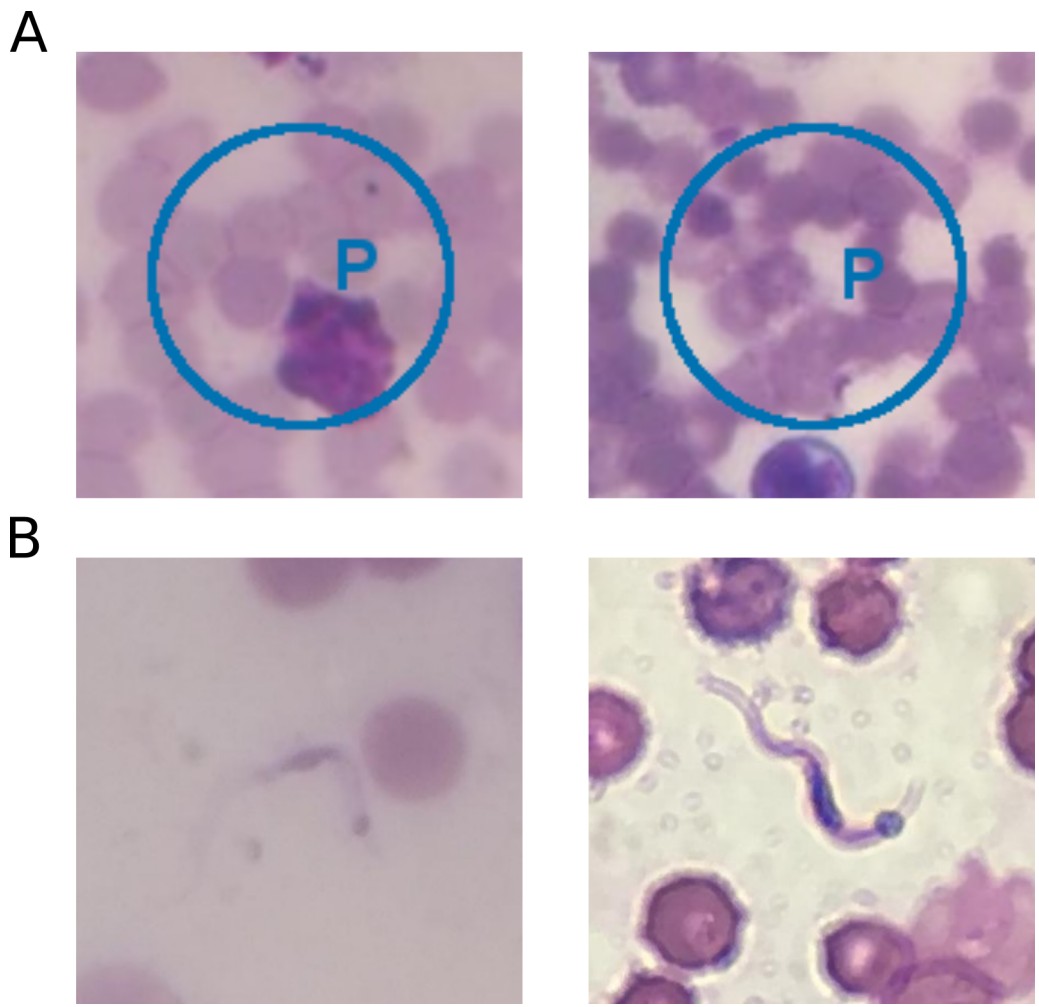

**Figure 6  Sample images of false-positive and false-negative Chagas parasite detection algorithm.** (A) The regions of the images with leukocytes (left) or high density of red cells (right) showed overstained areas that made it difficult to properly classify by the algorithm. (B) Parasites in regions of the image that presented low contrast (left) or low sharpness (right) were not recognized by the algorithm.

in this kind of study is image acquisition and ground truth annotation. To obtain and annotate this large number of images is particularly difficult, and it is more difficult to apply these techniques in a neglected disease context.

The mobile phone camera mainly affects the outline of objects in the image and the sharpness. The algorithm we present also has its results affected by such characteristics. A higher false-positive rate was observed in regions of the image with leukocytes or high red cell density. A higher false-negative rate was observed in regions of low contrast and sharpness. Therefore, smear quality directly affects classifier performance. It is extremely important to reduce the false-negative rate, since undiagnosed patients can be left without proper treatment and in a life-threatening situation. We recommend evaluating the algorithm on images acquired from samples with different staining time and
**Table 6 Comparison of the results of our algorithm with other published studies.**

| Reference | Parasite | Image capture device | ML Model | Sensitivity (%) | Specifity (%) |
|---|---|---|---|---|---|
| The present work | *T. cruzi* | Mobile phone camera | RF | 90.5 | 88.6 |
| *Uc-Cetina, Brito-Loeza & Ruiz-Piña (2013)* | *T. cruzi* | Dedicated camera | Gaussian discriminant | 98.3 | 84.4 |
| *Uc-Cetina, Brito-Loeza & Ruiz-Piña (2015)* | *T. cruzi* | Dedicated camera | AdaBoost + SVM | 100 | 93.2 |
| *Pereira et al. (2020)* | *T. cruzi* | Dedicated camera | Convolutional Neural Network | 97.6 | 95.2 |
| *Soberanis-Mukul et al. (2013)* | *T. cruzi* | Dedicated camera | KNN | 98 | 85 |
| *Savkare & Narote (2012)* | *Plasmodium* spp. | Dedicated camera | SVM | 96.3 | 99.1 |
| *Yang et al. (2020)* | *Plasmodium* spp. | Mobile phone camera | Convolutional Neural Network | 92.6 | 94.3 |
| *Rajaraman et al. (2018)* | *Plasmodium* spp. | Mobile phone camera | Convolutional Neural Network | 94.5 | 96.9 |
| *Rosado et al. (2016)* | *Plasmodium* spp. | Mobile phone camera | SVM | 80.5 | 93.8 |
| *Oliveira et al. (2017)* | *Plasmodium* spp. | Mobile phone camera | Adaboost | 59 | 95 |

**Notes.**

RF, random forest; SVM, support vector machine; KNN, k-nearest neighbours.

dye concentration. Such an assessment can further validate the robustness of the algorithm and identify optimal sample preparation.

In summary, our results demonstrate that the proposed algorithm can detect trypomastigote forms of *T. cruzi* in images acquired with a mobile device attached to a microscope. Automating image analysis acquired with a mobile device is a viable alternative for reducing costs and gaining efficiency in the use of the optical microscope. We hope that this algorithm can serve as a tool for early diagnosis of Chagas disease.

### Funding

This work was supported by the São Paulo Research Foundation (FAPESP, grant numbers 2020/12017-9 to Mauro César Cafundó Morais, 2018/14933-2 to Helder Nakaya, 2015/22308 to Luciano da F. Costa) and National Council for Research (CNPq grant n. 307085/2018-0). The funders had no role in study design, data collection and analysis, decision to publish, or preparation of the manuscript.

### Grant Disclosures

The following grant information was disclosed by the authors:
São Paulo Research Foundation (FAPESP): 2020/12017-9, 2018/14933-2, 2015/22308.
National Council for Research (CNPq): 307085/2018-0.

### Competing Interests

Helder I. Nakaya is an Academic Editor for PeerJ.

### Author Contributions

- Mauro César Cafundó Morais conceived and designed the experiments, performed the experiments, analyzed the data, prepared figures and/or tables, authored or reviewed drafts of the paper, and approved the final draft.
- Diogo Silva conceived and designed the experiments, performed the experiments, analyzed the data, prepared figures and/or tables, authored or reviewed drafts of the paper, and approved the final draft.
- Matheus Marques Milagre conceived and designed the experiments, performed the experiments, analyzed the data, authored or reviewed drafts of the paper, and approved the final draft.
- Maykon Tavares de Oliveira conceived and designed the experiments, performed the experiments, authored or reviewed drafts of the paper, and approved the final draft.
- Thaís Pereira performed the experiments, authored or reviewed drafts of the paper, and approved the final draft.
- João Santana Silva conceived and designed the experiments, analyzed the data, authored or reviewed drafts of the paper, and approved the final draft.
- Luciano da F. Costa conceived and designed the experiments, analyzed the data, authored or reviewed drafts of the paper, and approved the final draft.

- Paola Minoprio conceived and designed the experiments, analyzed the data, authored or reviewed drafts of the paper, and approved the final draft.
- Roberto Marcondes Cesar Junior conceived and designed the experiments, analyzed the data, authored or reviewed drafts of the paper, and approved the final draft.
- Ricardo Gazzinelli conceived and designed the experiments, authored or reviewed drafts of the paper, and approved the final draft.
- Marta de Lana conceived and designed the experiments, analyzed the data, authored or reviewed drafts of the paper, and approved the final draft.
- Helder I. Nakaya conceived and designed the experiments, analyzed the data, authored or reviewed drafts of the paper, and approved the final draft.

### Animal Ethics

The following information was supplied relating to ethical approvals (i.e., approving body and any reference numbers):

The Ethics Committee for the Use of Animals (CEUA) at the Federal University of Ouro Preto, Laboratory of Chagas Disease, Minas Gerais, Brazil provided approval for this research (CEUA No. 2015/50).

### Data Availability

The image data files are available at Zenodo: https://zenodo.org/record/5123062.

The image annotations and features extracted from the parasite's objects are available in the Supplementary Files.

The code used for model development are available at Bitbucket: https://bitbucket.org/dmatos88/jmire2/src/master/.

The code to test the algorithm is available at GitHub: https://github.com/csbl-br/chagas_detection/.

### Supplemental Information

Supplemental information for this article can be found online at http://dx.doi.org/10.7717/peerj.13470#supplemental-information.

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
