# Peer review of "Automatic detection of the parasite Trypanosoma cruzi in blood smears using a machine learning approach applied to mobile phone images"

_PeerJ, doi:10.7717/peerj.13470_

## Round 0.1 · original submission · Major Revisions

As you will see in the following sections, all four reviewers - to whom we are grateful for their dedication to reviewing your work - agreed that your manuscript requires revisions, either minor or major. Hence, I strongly suggest you proceed with a careful and thorough revision of all the raised issues. In particular, much attention should be paid to the methodological issues raised by reviewers 3 and 4, also including a more comprehensive a state-of-the-art revision that may compare your method with those available in the literature. On the other hand, as noted by the reviewers, it is also my concern that your manuscript requires extensive proofreading by a fluent English speaker.

Reviewer 1 ·

Basic reporting

The manuscript is clearly written and adequately exposes the study problem, however, it does present some lines that require a revision of the English language (eg lines 38-39, 78). The arguments presented are supported by specialized and updated references, however, some editorial observations are noted in the text (eg lines 38-39, 261-264, among others). The methodological approach and the results are adequately presented in images and tables that allow a better understanding to the reader.

Experimental design

The proposal of the work is to use the camera of a mobile phone to acquire the images and their subsequent analysis for the recognition of trypanosomes. The massive use of this electronic device can be used for different and varied purposes, such as this research. The methodological approach described was clear and would allow its replication without problem, although I consider that too many characteristics of the parasite were considered that could affect its results, such as the different categories that they established for the comparison of the performance of the algorithm (body, kinetoplast, body and kinetoplast, body and nucleus, and T. cruzi), although in the end the authors recognized it. The authors are advised to explore other variables in the treatment of the blood to improve its staining and recognition of the parasite (lines 339-341). Statistical analyzes were supported by pertinent literature. The manuscript is in the appropriate range for the objectives and interest of the journal

Validity of the findings

It is necessary for the authors to reinforce their arguments for the application of their results, since there are other algorithms and proposals that have shown better performance in the recognition of the parasite. For example, explain where your findings are located and how they can be applied. Otherwise, the results found appear to be irrelevant.
The discussion is well oriented based on the existing knowledge on the subject, and has a good self-criticism in the results. Recommendations are made to the authors for potential publications with a greater quantity and diversity of images.

Additional comments

The generation of knowledge through new approaches will always be welcome, in such a way that it allows to propose or design better strategies to solve different social problems, such as in this case, the diagnosis of Chagas disease. In this context, this manuscript fulfills this function in the generation of potentially applicable knowledge.

Annotated reviews are not available for download in order to protect the identity of reviewers who chose to remain anonymous.

Reviewer 2 ·

Basic reporting

This work describes an approach to automatically count trypanosomes in blood smear images, captured using a mobile phone, as a quick and accessible method of detecting infection. As identificaton of parasites in a blood smear is the gold standard for diagnosis this is a simple but valuable tool.

Overall the paper is suitably structured and presented. It is readable and seems to cite literature well. There is one major problem with the text:

Use of "dpi" to describe resolution does not make sense. How are standard print resolutions related at all to image pixel sizes in the microscopy system? Resolution is also probably not the correct term, this presumably refers to pixel size rather than the resolving power of the optical system.
Please list instead the relevant parameters about 1) the typical scale of the images in question, in units of micrometres per pixel, and 2) the dimensions of the images in question, in units of pixels or micrometres. The overall optical system should also be described, particularly objective lens magnification and numerical aperture, any additional magnification (eyepiece or camera mount) and the physical pixel size in the cameras used/referred to. That allows calculation of actual resolving power. Only then can the mobile phone images really be compared to dedicated microscope camera images.
This affects many places in the text, line 43, 49, 99, 102, 109, etc. etc.

I found it hard to understand much of the detail around the actual images going into the study due to this poor description - this needs to be corrected carefully. Overall, I get the impression that image pixel is probably not the limiting factor at all - but a combination of contrast and actual resolution in comparison to using a much larger and more expensive microscope and dedicated camera.

Paragraph starting line 237:
This is the first description of the images that makes sense. Note again, "resolution" in microscopy systems normally refers to the resolving power (the objective type is not specified, but for a 100x oil immersion objective resolution is ~0.2um). 2.835 pixels/um is the image's scale.

Experimental design

Paragraph starting 343:
I'd particularly like to see performance of your method on images captured by a dedicated microscope camera rather than a mobile phone - that would be the 'gold standard' for comparison to this approach.

Paragraph starting 254:
Does perceptual lightness for thresholding help with thresholding the image?
How can classes 1) and 3) occur - kinetoplast or nucleus without a cell body? I found this description unclear. Surely this reflects a limitation of the original intensity thresholding?
It seems that the limiting factor in this approach will be this intensity thresholding, although overall the method seems to works.

Paragraph starting 305:
The miss-clasification of leukocytes seems like a serious limitation, especially as leukocyte number can be affected by many different conditions. Could they be included as a category for the training to allow them to be positively identified rather than a false positive?

Validity of the findings

I suspect the classification is probably quite over-fitted to the specific parameters of the combination of this particular sample preparation and the particular combination of mobile phone and microscope. It seems to have only been tested with one microscope and camera, and one sample? Performance on some number of different samples, prepared on different days, by different researchers and then tested on different microscopes with different mobile phone cameras should really be tested to convince me that this tool could actually be used by other research groups.

The results shown from this dataset appear robust. However, this does not mean this tool is robust. As such, currently it is an interesting proof-of-concept, but there is not evidence that it is a usable tool - the discussion should reflect this.

Additional comments

Paragraph starting line 237:
Is there such a thing as an uncompressed JPEG? I thought the chroma data is always downsampled and the storage of the frequency domain representation of the luminance is always at least a little compressed.

Paragraph starting 283:
It is unclear what execution this time refers to. Training or classification? Indicating that it is (presumably) a purely CPU workload would be useful.

Paragraph starting 305:
Issues with out-of-focus false negatives could likely be solved by cropping the field of view. Is the 'out-of-focus' referred to here truly focus problems, is it an issue with focal plane planarity? or chromatic aberration?

Paragraph starting 353:
As Plasmodium is an intracellular parasite, in comparison to the extracellular blood stage of T. cruzi, this comparison doesn't seem 'fair' - some consideration of the differing biology is needed.

Paragraph starting 361:
It is still unclear to me what the limiting factor of the mobile device camera is here. It is presumably not actually resolution of the microscope

Overall comments:
Overall this is a potentially valuable study. The ability to automatically quantify cruzi parasitaemia using a mobile phone and a standard portable 'student'-type microscope is a genuinely useful approach. As I indicate above, I have significant concerns that it would be a useable tool, but stands as an interesting proof-of-concept.

I must note that I find the overall approach a little surprising. Surely an advanced object detection algorithm like YOLO (https://pjreddie.com/darknet/yolo/) could perform significantly better? However, this does not invalidate the results shown.

Reviewer 3 ·

Basic reporting

In general, applications of machine learning to biomedical problems are always welcome and, this article proposes a method on this line. However, I think that the paper needs reworking to clarify the main ideas. For instance, it is not clear how the classifier is used with six classes. SVM is not a multi-class classifier therefore, you should report which of the known strategies to deal with this was used in this work. An illustration with the pipeline of the whole algorithm/method would be welcome.

In Figure 1, the ROC curve is hard to read.

Experimental design

According to the paper, the feature extraction was performed over the segmented image. This to me, seems to be very inefficient since the segmentation algorithm is a very simple one and for sure there must be plenty of inaccurately segmented parasites. Doing manual segmentation of the kinetoplast and nucleus and the feature extraction would be a much better idea.

You mention that PCA was performed and only the principal K components were used for training. What is the value of K?

In the object classification section, you should introduce your classes and no later. On the contrary, the paragraph in this section is incomprehensible.

The manual labeling section is not clear. You describe that manually by a point-and-click event you are labeling the nucleus and kinetoplast of each parasite. Therefore, you are not finding the centroid of each region of interest (ROI), nor creating a bounding box around them, nor labeling each pixel of the ROIs. What is the point to have an almost randomly selected coordinated inside each ROI?

I also don’t understand what you mean by the field-of-view identifier, the pointed object
(Nucleus or kinetoplast). (Lines 250 to 253)

Then, you introduce 6 different classes. First, I don’t see the point in doing that because you only have features for the kinetoplast and nucleus. Therefore, it is natural to wonder how you are representing the other four classes.

In line 283, you mention “We observed a performance gain after feature selection”. Are you referring to gains (reduction) in processing time? Is this processing time only for PCA or the whole algorithm? Later you will state that PCA reduction did not work together with the classifier. Then, why give times of something that did not work?

Then you report your classification results, showing that PCA did not help at all. Since you did not report the number of PCA components used (the value of K) it is difficult to comment about that.

The same comment applies to the definition of your classes. Why use so many if you only want to detect the parasite? Introducing many classes makes the classifier struggle and the paper difficult to read.

The problems you are reporting with respect to objects of similar color are expected. Including color in the features maybe not be a good idea. The stain may vary due to many reasons and therefore the algorithm is prone to errors.

Validity of the findings

My other main concern is that the reported results are below some published works. I guess this is because a simple method such as SVM was used and no other techniques such as Adaboost, neural networks and so on that may boost the results. It seems then that the only novelty here is the use of low-resolution images.

Reviewer 4 ·

Basic reporting

Tables and figures must be resized so that they fall on the same page with the title (empty pages should be avoided).

Experimental design

In this paper, the authors present classification results of T. cruzi trypomastigotes in low-resolution images (72 dpi) using SVM based algorithm for the detection. To correctly categorize a candidate parasite, the proposed method needed the selection of a non-fixed threshold value involving segmentation before classification. Images were converted to CIEL*a*b* color space, and the objects of interest were segmented with the adaptive (local) threshold technique. Whereas for the classification they marked the position of each nucleus and kinetoplast of the trypomastigotes found in the image and constructed 6 classes: 1) kinetoplast; 2) kinetoplast with cell body; 3) nucleus; 4) nucleus with cell body; 5) T. cruzi; 6) unknown.
The model was capable of recognizing the parasites in the training set with 91.4% and 91.7% accuracy and sensitivity, respectively, and with a sensitivity of 89.6% in a set of test set data.

The overall work has a lot of question marks:
1) The authors claim that the model works on low-resolution images (72 dpi) where we don't see a comparison with the existing models where the data is collected as high-resolution. For such data sets, the authors can try their model and show the comparison with the existing models. The other way around, the authors can use the existing machine learning models on the newly collected data set and apply them by resetting the train and validation set. In this way, the authors can claim that the model they provide is better for low-resolution images.
The authors can use for such comparison already cited works (such as Uc-Cetin work ) or the work of other works which are not cited such as

da Silva Pereira, Andr´e, Alexandre dos Santos Pyrrho, Daniel Figueiredo Vanzan, Leonardo Oliveira Mazza, and Jos´e Gabriel RC Gomes. ”Deep Convolutional Neural Network applied to Chagas Disease Parasitemia Assessment.”Congresso Brasileiro de Inteligˆencia Computacional, January 2020 , DOI:10.21528/CBIC2019-119

2) I see that the authors adaptive (local) threshold technique. This technique will lead to noise sensitivity. Did the authors take this into account? If so would be good to be further explained. The proposed method needs a better description with sufficient detail on all the other image processing techniques and their parameters limitation.

3) The features generation of 6 classes is confusing. Why the authors wouldn't use a binary classifier for automated T. cruzi detection? I am afraid such a classification leads to an increase in the accuracy shown in the paper.

4) Górriz et al., 2018 is not found as reference.

Considering the above-mentioned comments I recommend the authors resubmit the paper with a major revision.

Validity of the findings

To be considered in case the authors answer the comments in the Experimental design.

Additional comments

no additional comments

---

## Round 0.2 · Minor Revisions

Despite that the current version of the article is certainly improved with respect to the previous one, there are still some issues that needs to be addressed properly. As you will read below, two of three reviewers concluded that your work is ready for publication. However, reviewer 3 raised a set of issues that need to be addressed. In particular, reviewer 3 insisted in the unbalanced nature of the training dataset derived from the segmentation method. As you may certainly agree, using an unbalanced dataset to train any ML/AI method may derive in a heavily biased algorithm. Even achieving a high performance using a ROC curve, it may fail on generalization. Therefore, I agree with the reviewer on the need to improve the segmentation method. Importantly, even using a slow and error-prone manual procedure, if the labeling method results in a well-balanced dataset, this may not only useful for the purpose of this work, but also as a general tool to be further used by the academic community. On the other hand, in spite of the temptation to include results denoting the vastness of the work, I agree with reviewer 3 that there is no need to include PCA and CFS results. However, you should include these results in the supplementary information section. Finally, I also agree with the reviewer that using ensembles may improve the performance of the classifications made by the SVMs. I also encourage you to evaluate the contribution of ensembles to the classification performance.

Reviewer 1 ·

Basic reporting

I agree with the corrections made by the authors

Experimental design

The authors adequately addressed the observations made

Validity of the findings

The authors adequately addressed the observations made

Additional comments

I recommend that the authors continue to improve their device to achieve a contribution that can be applied to the human population

Reviewer 3 ·

Basic reporting

In my opinion, this article is not ready for publication yet. It is promising but some improvements need to be done.

There are some issues that I describe below.

Experimental design

The adaptive thresholding segmentation process is so simple that it is practically impossible to segment the whole parasite body. In Table 1, we can see that you ended up with only 111 T. Cruzi parasites in the training set and 23 in the test set. Meanwhile, the kinetoplast, nucleus, and unknown classes have a size in the order of thousands. Your dataset is quite unbalanced. Therefore, you either upgrade your segmentation method, use a manual segmentation method (at least for training) or apply any technique to deal with an unbalanced dataset (such as image augmentation).

Again, seems to me that the output of the segmentation algorithm forced you to go for six classes when you really should aim for two. Too many classes should be avoided when doing binary segmentation even more in unbalanced sets. Otherwise, the performance of the classification algorithm gets decimated.

Let me insist that I do not see the need to present the PCA and CFS algorithms since they did not improve the classification results. I suggest removing them and just mentioning in the manuscript that they were tested but delivered bad results.

The output from the SVM classification algorithm may be improved by using techniques such as ensemble methods. I encourage the authors to explore this possibility.

Validity of the findings

The performance of your proposal is far from close to one of the other techniques already published. The argument that this method is suitable for low-resolution images and the others not, is from my perspective, weak to say the least.

Reviewer 4 ·

Basic reporting

The revised version is acceptable for publication.

Experimental design

Fully valid.

Validity of the findings

Acceptable.

---

## Round 0.3 · accepted · Accept

Dear Dr. Morais and Dr. Nakaya.

After carefully reviewing your manuscript I'm glad to realize how much it has improved from the first iteration. In its current version, it is not only better organized but also more scientifically sound. Now your contributions appear clearer. Indeed, as requested by the reviewer, the inclusion of the new segmentation approach offers an interesting workaround to surpass the previous painstaking bottleneck behind your proposal. The unsupervised graph-based approach appears to be both fast and reliable. Despite a comparison with other existing alternatives, such as a tessellation process, would have certainly enriched your contribution, as implemented, im my opinion, it achieves the requirements from the reviewer. I wonder how much should improve the performance of your algorithm by using high resolution images? Regarding the class imbalance issue, the methodological improvement presented in this version of the manuscript seems to satisfy the issue raised by the reviewer. However, a more detailed analysis comparing the distribution of the previous dataset with the newer one, would have certainly enriched your work.

On a final note, I’d like to thank you for your patience, taking this opportunity to congratulate you and your team for the acceptance of this manuscript. It is an interesting contribution, with particular applications in the field.